# Resonant catalysis of thermally activated chemical reactions with vibrational polaritons

Jorge A. Campos-Gonzalez-Angulo [1], Raphael F. Ribeiro[1] & Joel Yuen-Zhou[1]*

Interaction between light and matter results in new quantum states whose energetics can modify chemical kinetics. In the regime of ensemble vibrational strong coupling (VSC), a macroscopic number $N$ of molecular transitions couple to each resonant cavity mode, yielding two hybrid light–matter (polariton) modes and a reservoir of $N-1$ dark states whose chemical dynamics are essentially those of the bare molecules. This fact is seemingly in opposition to the recently reported modification of thermally activated ground electronic state reactions under VSC. Here we provide a VSC Marcus–Levich–Jortner electron transfer model that potentially addresses this paradox: although entropy favors the transit through dark-state channels, the chemical kinetics can be dictated by a few polaritonic channels with smaller activation energies. The effects of catalytic VSC are maximal at light–matter resonance, in agreement with experimental observations.

---

[1] Department of Chemistry and Biochemistry, University of California San Diego, La Jolla, California 92093, USA. *email: joelyuen@ucsd.edu

The strong interaction between excitations in a material medium and a resonant confined electromagnetic mode results in new states with light–matter hybrid character (polaritons)[1,2]. Recent studies of molecular polaritons have revealed new phenomena and features that are appealing for applications in chemistry and materials science. These discoveries opened the doors to the emerging field of polariton chemistry[3–11]. Of particular interest are recent observations of chemoselective suppression and enhancement of reactive pathways for molecules whose high-frequency vibrational modes are strongly coupled to infrared optical cavities[12–15]. These effects of vibrational strong coupling (VSC) are noteworthy in that they occur in the absence of external photon pumping, implying that they involve thermally activated (TA) processes and potentially paving the road for a radically new synthetic chemistry strategy that involves injecting microfluidic solutions in suitable optical cavities (Fig. 1) to induce desired transformations. It is important to highlight that the VSC in these samples is the consequence of an ensemble effect: each cavity mode (i.e., resonant with the polarization of the material) coherently couples to a large number of molecules. This coupling leads to two polaritonic modes and a macroscopic set of quasi-degenerate dark (subradiant) modes that, to a good approximation, should feature chemical dynamics that is indistinguishable from that of the bare molecular modes[16]. This picture could potentially change as a consequence of ultrastrong coupling effects; however, these effects should not be significant for modest Rabi splittings as those observed in the experiments[12–15].

Of the population of vibrationally excited states at thermal equilibrium, a tiny fraction would be allocated to the polariton modes, with the overwhelming majority residing in the dark-state reservoir[17–20], unless the temperature is low enough for the lower polariton to overtake the predominant population second to that of the ground state. It is thus puzzling and remarkable that differences in the chemical kinetics can be detected in macroscopic systems under VSC at room temperature. This study provides a possible rationale for these observations. By studying a VSC version of the well-established Marcus–Levich–Jortner (MLJ) TA electron transfer model[21–23], we find a parameter range where, even if the number of dark-state channels massively outweigh the few polaritonic ones, the latter dictate the kinetics of the reaction given their smaller activation energies. The present model does not feature the complexity of the experimentally studied systems; however, it provides a minimalistic conceptual framework to develop qualitative insights on general TA VSC processes. We believe that this mechanism of polaritonic activation barrier reduction might be a widespread feature among such processes.

## Results

### Theoretical framework.
According to the MLJ theory, the rate coefficient of charge transfer from a reactant (R) to a product (P) electronic state, at constant temperature $T$, is given by[21–23]

$$k_{R \to P} = \sqrt{\frac{\pi}{\lambda_S k_B T}} \frac{|J_{RP}|^2}{\hbar} e^{-S} \times \sum_{\nu=0}^{\infty} \frac{S^\nu}{\nu!} \exp\left(-\frac{(\Delta E + \lambda_S + \nu\hbar\omega_P)^2}{4\lambda_S k_B T}\right),$$

(1)

where $J_{RP}$ is the non-adiabatic coupling between electronic states, $\lambda_S$ is the outer-sphere reorganization energy related to the low-frequency (classical) degrees of freedom of the solvent, $\omega_P$ is the frequency of a high-frequency intramolecular (quantum) mode with quantum number labeled by $\nu$, $S = \lambda_P/\hbar\omega_P$ is a Huang–Rhys parameter with $\lambda_P$ the reorganization energy of the quantum mode, $\Delta E$ is the difference in energy between the equilibrium configurations of the R and P potential energy surfaces, and $k_B$ is the Boltzmann constant. The MLJ rate can be thought of as a generalization of Marcus theory to include a sum over channels

with different quanta $\nu$ in the high-frequency mode of the product.

To gauge the effects of VSC, we consider the interaction between a single microcavity mode and an ensemble of $M$ molecules that undergo electron transfer. For simplicity, we assume that VSC occurs via the high-frequency mode of P (as the MLJ rate only accounts for transitions originated in the ground state of the reactants, the case where this coupling also happens through R shares features with the current one that we shall discuss later). This constraint implies a drastic change in molecular geometry upon charge transfer so that the vibrational transition dipole moment goes from negligible to perceptible. This rather unusual behavior can be observed in molecular actuators[24,25]. The Hamiltonian for such system is

$$\hat{H} = \hat{H}_{ph} + \sum_{i=1}^{M}\left[\hat{H}_R^{(i)}|R_i\rangle\langle R_i| + \left(\hat{H}_P^{(i)} + \hat{\mathcal{V}}_{int}^{(i)}\right)|P_i\rangle\langle P_i| + J_{RP}(|R_i\rangle\langle P_i| + |P_i\rangle\langle R_i|)\right],$$

(2)

where $\hat{H}_{ph} = \hbar\omega_0\left(\hat{a}_0^\dagger\hat{a}_0 + \frac{1}{2}\right)$ is the Hamiltonian of the electro-magnetic mode with frequency $\omega_0$, $|R_i\rangle$ and $|P_i\rangle$ denote the electronic (reactant/product) states of the $i$-th molecule, $\hat{H}_R^{(i)} = \hbar\omega_R\hat{\mathcal{D}}_i^\dagger\hat{\mathcal{S}}_i^\dagger\left(\hat{a}_i^\dagger\hat{a}_i + \frac{1}{2}\right)\hat{\mathcal{S}}_i\hat{\mathcal{D}}_i + \hat{H}_S(\hat{q}_S^{(i)} + d_S)$ and $\hat{H}_P^{(i)} = \hbar\omega_P\left(\hat{a}_i^\dagger\hat{a}_i + \frac{1}{2}\right) + \hat{H}_S(\hat{q}_S^{(i)}) + \Delta E$ are the bare Hamiltonians of the $i$-th reactant/product with quantum mode frequency $\omega_R$ and $\omega_P$, respectively. $\hat{\mathcal{V}}_{int}^{(i)} = \hbar g\left(\hat{a}_i^\dagger\hat{a}_0 + \hat{a}_0^\dagger\hat{a}_i\right)$ is the light–matter interaction under the rotating wave approximation[26] with single-molecule coupling $g = -\mu\sqrt{\frac{\hbar\omega_0}{2V\epsilon_0}}$, transition dipole moment $\mu$, and cavity mode volume $V$, $\hat{a}_i^\dagger/\hat{a}_i$ are creation/annihilation operators acting on the quantum mode of the $i$-th molecule ($i = 0$ denotes the cavity mode), $\hat{\mathcal{S}}_i = \exp\left[\frac{1}{2}\ln\left(\sqrt{\frac{\omega_P}{\omega_R}}\right)(\hat{a}_i^{\dagger 2} - \hat{a}_i^2)\right]$ and $\hat{\mathcal{D}}_i = \exp\left[\frac{1}{\sqrt{2}}(\hat{a}_i^\dagger - \hat{a}_i)d_P\right]$ are squeezing and displacement operators[26] (see Supplementary Note 1 for the origin of these terms), $\hat{H}_S(\hat{q}_S^{(i)}) = \sum_\ell \frac{1}{2}\hbar\omega_S^{(\ell)}\left(\hat{p}_S^{(i,\ell)2} + \hat{q}_S^{(i,\ell)2}\right)$ is the Hamiltonian of the classical modes with frequencies $\omega_S^{(\ell)}$, $\hat{p}_S^{(i)}$ and $\hat{q}_S^{(i)}$ are the set of rescaled low-frequency momenta and positions associated with the $i$-th quantum mode, and $d_P$ and $\mathbf{d}_S$ are the rescaled (dimensionless) distances between equilibrium configurations of the reactant and product along the quantum and classical mode coordinates, respectively. We shall point out that, as it only considers coupling to a single cavity mode, the Hamiltonian in Eq. (2) entails coarse-graining; therefore, $M$ is not the total number of molecules in the cavity volume, but the average number of molecules coupled per cavity mode[17]. Although polaritonic effects in electron transfer processes have been studied in the pioneering work of ref. [27] (see also ref. [28]), we note that they were considered in the electronic strong coupling regime; as we shall see, the vibrational counterpart demands a different formalism and offers conceptually different phenomenology.

As a consequence of VSC, the system is best described in terms of collective normal modes defined by the operators[7,29]

$$\hat{a}_{+(N)} = \cos\theta_N\hat{a}_0 - \sin\theta_N\hat{a}_{B(N)},$$
$$\hat{a}_{-(N)} = \sin\theta_N\hat{a}_0 + \cos\theta_N\hat{a}_{B(N)},$$
$$\hat{a}_{D(N)}^{(k)} = \sum_{i=1}^{N} c_{ki}\hat{a}_i; \quad 2 \leq k \leq N$$

(3)

where $0 \leq N \leq M$ is the number of molecules in the $P$ state at a given stage in the reaction. These operators correspond to the

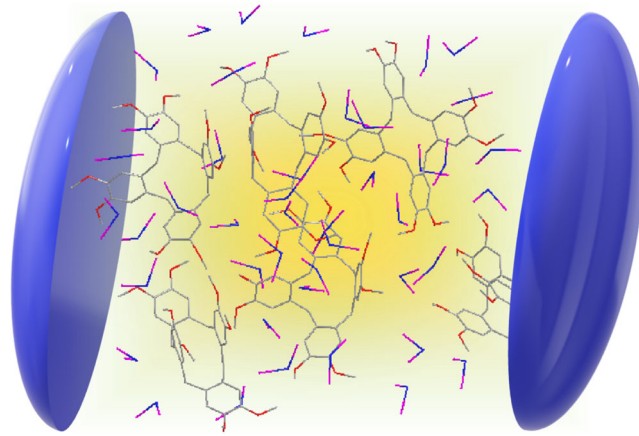

**Fig. 1** Depiction of a microcavity. A large number of molecules can undergo a chemical reaction (e.g., electron-transfer-induced conformational transformation[24]) and support a high-frequency vibrational mode that can strongly couple to a confined optical mode; these molecules are in a solvated environment (blue/purple moieties). The reaction of concern is mediated by that intramolecular mode and a collection of low-frequency modes of the solvation sphere. The optical mode is typically confined by two dielectric mirrors (blue structures) separated by a spacer that is saturated with the reaction mixture

upper and lower polaritons (UP and LP), and dark (D) modes, respectively. It is noteworthy that the operators $\hat{a}_{\mathrm{D}(N)}^{(k)}$ are defined only for $N \geq 2$ and the coefficients $c_{ki}$ fulfill $\sum_{i=1}^{N} c_{ki} = 0$ and $\sum_{i=1}^{N} c_{k'i}^{*} c_{ki} = \delta_{k'k}$. In Eq. (3), $\theta_N = \frac{1}{2} \arctan \frac{2g\sqrt{N}}{\Delta}$ is the mixing angle, where $\Delta = \omega_0 - \omega_{\mathrm{P}}$ is the light–matter detuning and $\hat{a}_{\mathrm{B}(N)} = \frac{1}{\sqrt{N}} \sum_{i=1}^{N} \hat{a}_i$ corresponds to the so-called bright (super-radiant) mode. These modes have associated frequencies

$$\omega_{\pm(N)} = \frac{\omega_0 + \omega_{\mathrm{P}}}{2} \pm \frac{\Omega_N}{2},$$
$$\omega_{\mathrm{D}} = \omega_{\mathrm{P}}, \qquad (4)$$

where $\Omega_N = \sqrt{4g^2 N + \Delta^2}$ is the effective Rabi splitting; equivalent definitions can be made for the creation operators. It is worth noting that there is no free lunch: the superradiantly enhanced VSC with the bright mode occurs at the expense of the creation of a macroscopic number of dark modes that—under the context of this model—do not mix with light (inhomogeneous broadening results in small but experimentally observable light-like character for these modes[29–32]). This effect is negligible for the phenomena considered in this work given that the density of molecular excitations is much larger than that of the photon modes.)

Inside of the cavity, the reaction $\mathrm{R} \longrightarrow \mathrm{P}$ becomes

$$\mathrm{R} + \mathrm{UP}_{N-1} + \mathrm{LP}_{N-1} + \sum_{k=2}^{N-1} \mathrm{D}_{N-1}^{(k)} \longrightarrow \mathrm{UP}_N + \mathrm{LP}_N + \sum_{k=2}^{N} \mathrm{D}_N^{(k)}, \qquad (5)$$

where the subscripts indicate the number of molecules that participate in VSC (from Eq. (3), it can be seen that $\mathrm{UP}_0$ corresponds to the uncoupled photon mode, and $\mathrm{LP}_0$ and $\mathrm{D}_0^{(k)}$ are non-existent). This reaction implies that each time a molecule transforms into the product, it becomes part of the ensemble that couples to light (see Supplementary Note 2 for additional insight). Electron transfer occurs as a result of a vibronic transition between diabatic states; this feature makes it similar to Raman scattering. A study of the latter under VSC[33] took advantage of

the massive degeneracy of the dark modes to introduce a judicious basis[34],

$$\hat{a}_{\mathrm{D}}^{(k)} = \frac{1}{\sqrt{k(k-1)}} \left( \sum_{i=1}^{k-1} \hat{a}_i - (k-1)\hat{a}_k \right), \qquad (6)$$

that enables calculations for an arbitrary number of molecules and will prove to be convenient for our purposes. Notice that the mode $\hat{a}_{\mathrm{D}}^{(k)}$ is highly localized at $\hat{a}_k$ but has a long tail for $\hat{a}_{1 \leq i \leq k-1}$ (for a visualization, see Supplementary Fig. 1); furthermore, it is fully characterized by the index $k$ and thus does not depend explicitly on $N$. In terms of these dark modes, the reaction in Eq. (5) can be drastically simplified from an $N+1$ to a three-body process,

$$\mathrm{R} + \mathrm{UP}_{N-1} + \mathrm{LP}_{N-1} \longrightarrow \mathrm{UP}_N + \mathrm{LP}_N + \mathrm{D}_N^{(N)}, \qquad (7)$$

where, without loss of generality, we have considered that the $N$-th molecule is the one that undergoes the reaction (notice that, in accordance with the notation introduced in Eq. (6), the mode $\mathrm{D}_N^{(N)}$ is highly localized in $P_N$ for sufficiently large $N$). Furthermore, we can identify the photon ($\hat{a}_0$), the $N$-th molecule ($\hat{a}_N$), and the bright state that excludes it ($\hat{a}_{\mathrm{B}(N-1)}$) as the normal modes embodying the natural degrees of freedom of the problem, since the modes in reactants and products can be written as Duschinsky transformations[35] of these. Explicitly, for the reactants we have

$$\begin{pmatrix} \hat{a}_{+(N-1)} \\ \hat{a}_{-(N-1)} \end{pmatrix} = \begin{pmatrix} \cos\theta_{N-1} & -\sin\theta_{N-1} \\ \sin\theta_{N-1} & \cos\theta_{N-1} \end{pmatrix} \begin{pmatrix} \hat{a}_0 \\ \hat{a}_{\mathrm{B}(N-1)} \end{pmatrix}, \qquad (8)$$

$$\hat{a}'_N = \hat{D}_N^{\dagger} \hat{S}_N^{\dagger} \hat{a}_N \hat{S}_N \hat{D}_N, \qquad (9)$$

where $\hat{a}'_N$ acts on the vibrational degrees of freedom of the $N$-th reactant (see Supplementary Note 1 for a derivation), whereas for the products

$$\begin{pmatrix} \hat{a}_{+(N)} \\ \hat{a}_{-(N)} \\ \hat{a}_{\mathrm{D}}^{(N)} \end{pmatrix} = \begin{pmatrix} \cos\theta_N & -\sin\theta_N & 0 \\ \sin\theta_N & \cos\theta_N & 0 \\ 0 & 0 & 1 \end{pmatrix} \begin{pmatrix} 1 & 0 & 0 \\ 0 & \sqrt{\frac{N-1}{N}} & \sqrt{\frac{1}{N}} \\ 0 & \sqrt{\frac{1}{N}} & -\sqrt{\frac{N-1}{N}} \end{pmatrix} \begin{pmatrix} \hat{a}_0 \\ \hat{a}_{\mathrm{B}(N-1)} \\ \hat{a}_N \end{pmatrix}. \qquad (10)$$

With the above considerations, the VSC analog of the MLJ rate coefficient in Eq. (1) is given by a sum over possible quanta $\{v_+, v_-, v_{\mathrm{D}}\}$ in the product modes $\mathrm{UP}_N$, $\mathrm{LP}_N$, and $\mathrm{D}_N^{(N)}$, respectively:

$$k_{\mathrm{R}\to\mathrm{P}}^{\mathrm{VSC}} = \sqrt{\frac{\pi}{\lambda_{\mathrm{S}} k_{\mathrm{B}} T}} \frac{|J_{\mathrm{RP}}|^2}{\hbar} \sum_{v_+=0}^{\infty} \sum_{v_-=0}^{\infty} \sum_{v_{\mathrm{D}}=0}^{\infty} W_{v_+,v_-,v_{\mathrm{D}}}, \qquad (11)$$

where $W_{v_+,v_-,v_{\mathrm{D}}} = \left| F_{v_+,v_-,v_{\mathrm{D}}} \right|^2 \exp\left( -\frac{E_{v_+,v_-,v_{\mathrm{D}}}^{\ddagger}}{k_{\mathrm{B}} T} \right)$, and

$$\begin{aligned} \left| F_{v_+,v_-,v_{\mathrm{D}}} \right|^2 &= \left| \left\langle 0_{+(N-1)} 0_{-(N-1)} 0_{\mathrm{R}} | v_+ v_- v_{\mathrm{D}} \right\rangle \right|^2 \\ &= \left( \frac{\sin^2\theta_N}{N} \right)^{v_+} \left( \frac{\cos^2\theta_N}{N} \right)^{v_-} \left( \frac{N-1}{N} \right)^{v_{\mathrm{D}}} \\ &\quad \times \binom{v_+ + v_- + v_{\mathrm{D}}}{v_+, v_-, v_{\mathrm{D}}} \left| \left\langle 0' | v_+ + v_- + v_{\mathrm{D}} \right\rangle \right|^2, \end{aligned} \qquad (12)$$

is a Franck–Condon factor between the global ground state in the reactants and the excited vibrational configuration in the product[33]. Here, $|0'\rangle$ is the vibrational ground state of the $N$-th molecule in the reactant electronic state and $|v_+ + v_- + v_{\mathrm{D}}\rangle$ is the vibrational state of the $N$-th molecule with $v_+ + v_- + v_{\mathrm{D}}$ in the product electronic state. The calculation in Eq. (12) (see Supplementary Note 3 for a derivation) is reminiscent to the contemporary problem of boson sampling[36]. Using the notation

from Eq. (1),

$$E^{\ddagger}_{\nu_+, \nu_-, \nu_D} = \frac{(E_P^{\nu_+, \nu_-, \nu_D} - E_R^0 + \lambda_S)^2}{4\lambda_S},\qquad(13)$$

is the activation energy of the channel, with $E_R^0 = \frac{\hbar}{2} \, (\omega_{+(N-1)} + \omega_{-(N-1)} + \omega_R)$ and $E_P^{\nu_+, \nu_-, \nu_D} = \Delta E + \hbar[\omega_{+(N)}(\nu_+ + \frac{1}{2}) + \omega_{-(N)}(\nu_- + \frac{1}{2}) + \omega_P(\nu_D + \frac{1}{2})]$. Eq. (12) affords a transparent physical interpretation: the state $|\nu_+ \nu_- \nu_D\rangle$ is accessed by creating $\nu_+ + \nu_- + \nu_D$ excitations in the high-frequency oscillator of the $N$-th product; there are $\binom{\nu_+ + \nu_- + \nu_D}{\nu_+, \nu_-, \nu_D}$, ways to do so; $\left(\frac{\sin^2\theta_N}{N}\right)$, $\left(\frac{\cos^2\theta_N}{N}\right)$, and $\left(\frac{N-1}{N}\right)$ are the projections of the product normal modes on the oscillator of the $N$-th product; these scalings are the same as those obtained in our studies on polariton-assisted energy transfer (PARET)[37].

**Conditions for rate enhancement.** When $\omega_R = \omega_P$ and $N \gg 1$, the expressions for the Franck–Condon factor and activation energy simplify to

$$\left|F_{\nu_+, \nu_-, \nu_D}\right|^2 = \frac{e^{-S}}{\nu_+! \nu_-! \nu_D!} \left(\frac{S\sin^2\theta}{N}\right)^{\nu_+} \left(\frac{S\cos^2\theta}{N}\right)^{\nu_-} S^{\nu_D},\qquad(14)$$

$$E^{\ddagger}_{\nu_+, \nu_-, \nu_D} = \frac{[\Delta E + \lambda_S + \hbar(\nu_+\omega_+ + \nu_-\omega_- + \nu_D\omega_P)]^2}{4\lambda_S},\qquad(15)$$

where we have dropped the dependence of angles and frequencies on $N$ for brevity. For most of the experiments that have achieved VSC[12,16,19,38–41], the number of molecules that take part in the coupling is between $N = 10^6$ and $10^{10}$ per cavity mode[17]. For such orders of magnitude, at first glance, Eq. (12) would suggest that the contribution from the dark modes dominates the rate, which, according to Eq. (14), is the same as the bare case (Eq. (1)) for $\nu_D = 1$ and $\nu_+ = \nu_- = 0$, i.e., if the polaritons are not employed in the reaction. In fact, this was the conclusion for PARET[37], where coupling the product to transitions to the cavity led to no change in energy transfer from reactant molecules. However, the TA processes in electron transfer kinetics offers a new dimension to the problem that PARET does not feature. Careful inspection of the expressions at hand hints to the existence of parameters $\Delta E$ and $\lambda_S$ for which changes in the activation energy for the polariton channels dominate the rate. To find those parameters, we need first that the contribution going to the first vibrational excitation outplay that between ground states, i.e., $W_{001} > W_{000}$, which implies

$$\frac{\lambda_P}{\hbar\omega_P} > \exp\left(\frac{\hbar\omega_P}{4\lambda_S k_B T}[2(\Delta E + \lambda_S) + \hbar\omega_P]\right).\qquad(16)$$

Next, if the contribution from the channel where the product is formed with an excitation in the LP mode ($\nu_- = 1$, and $\nu_+ = \nu_D = 0$) dominates, then $W_{010} > W_{001}$, which yields

$$\frac{N}{\cos^2\theta} < \exp\left(\frac{\hbar(\Omega_N - \Delta)}{4\lambda_S k_B T} \times \left[\Delta E + \lambda_S + \hbar\omega_P + \frac{\hbar(\Delta - \Omega_N)}{4}\right]\right).\qquad(17)$$

The region of parameters that satisfies these inequalities for room-temperature $(k_B T \approx 0.2\hbar\omega_P)$ and typical experimental VSC Rabi-splittings $\hbar\Omega(\approx 0.1\hbar\omega_P)$[12,40] is illustrated in Fig. 2.

The order of magnitude of the plotted $\Delta E$ values is reasonably standard for this kind of processes[42,43], which suggests the experimental feasibility of attaining these conditions.

The effect of the electromagnetic mode and the conditions for which the enhancement of the polaritonic coupling can be

achieved is illustrated in Fig. 3. We can understand this effect as follows: the reaction takes place as a multi-channel process consisting of an electronic transition from the reactant global ground state into the product electronic state dressed with high-frequency vibrational excitations. As shown in Fig. 3 and Supplementary Fig. 2, the channel between global ground states is in the Marcus inverted regime[44,45] and, given the small value of the classical reorganization energy, the activation energy is fairly high. On the other hand, the channel to the first excited manifold is in the normal regime with a much lower activation energy. Additionally, for the parameters within the space illustrated in Fig. 2 the decrease in activation energy for the channel with an excitation in the LP mode is enough to overcome the elevated multiplicity of the dark modes (Fig. 3 and Supplementary Fig. 2), and effectively catalyze the electron transfer process. In terms of the expression for the rate coefficient, even though the entropic pre-exponential factor of the D channel is $N - 1$ times larger than that of the LP channel, the latter is associated with a larger exponential factor (lower activation energy).

In Fig. 2, we also show the parameter space that produces polaritonic enhancement as a function the detuning $\Delta$. It can be noticed that the range of admissible values for the classical reorganization energy increases as the detuning becomes negative. This can be understood from the fact that, for negative detunings, the frequency of the photon is smaller than that of the vibrational high-frequency mode and, therefore, the activation energy to LP is lower than that corresponding to $D$, thus providing more flexibility for parameters to fulfill the inequalities in Eq. (17). However, we must remark that this effect disappears at sufficiently large detunings, as the matter character of the LP becomes negligible to effectively mediate the electron transfer.

**Simulation of modified kinetics.** The overall effect of the cavity in the charge-transfer kinetics is displayed in Fig. 4, where we show the ratio of the rate coefficients, calculated inside $(k_{R\to P})$ and outside $(k_{R\to P}^{VSC})$ of the cavity as a function of the collective coupling $g\sqrt{N}/\omega_P$, for several values of detuning. The bell-shaped curves reflect the fact that, as the Rabi splitting increases, the activation energy of the LP decreases, thus making this channel the most prominent one. This trend goes on until $E_{010}^{\ddagger} = 0$, where this LP channel goes from the normal Marcus regime to the inverted one, and the activation energy starts to increase with the coupling until this pathway is rendered insignificant as compared with the transition to the D manifold, giving rise to kinetics indistinguishable from the bare molecules. The observation that larger positive detunings require stronger coupling to reach the maximum ratio of rate coefficients is due to the fact that the larger the photon frequency, the larger the median frequency of the polariton modes; therefore, a more substantial coupling is required to lower the frequency of the LP mode, and thus the activation energy associated with this channel. In addition, the trend observed in the maxima, which decrease with the detuning, can be regarded as a consequence of the previous effect: the larger couplings required to reach the zero-energy barrier are achieved with more molecules; thus, the contribution of LP becomes less relevant than that of D, as can be seen from the pre-exponential factors. Finally, a peculiar result is the fact that the effect on the rate coefficient is more prominent in a range of few molecules for slightly negative detunings. This observation should not come as surprising since, as previously mentioned, under this condition, the LP mode has a substantially decreased activation energy; therefore, for as small as it is, the light–matter coupling is enough to open a very favored channel that accelerates the reaction. This effect might end up quenched by dissipation; however, even in the

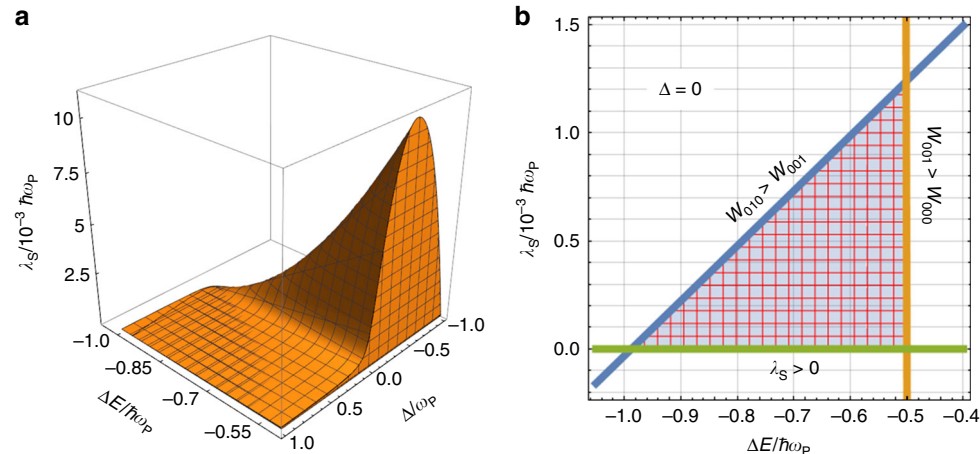

**Fig. 2** Electron transfer parameters for catalytic behavior. The lower polariton (LP) channel dominates the kinetics over the many dark (D) channels for these values of $\Delta E$ (the energy difference between product P and reactant R), $\lambda_S$ (the classical reorganization energy), and $\Delta$ (the detuning between the cavity and the high-frequency mode of the product). In **a** we explore the three variables, whereas in **b** we show the cross-section under the resonant condition. For these calculations, the reactant and product modes have equal high-frequencies $\omega_R = \omega_P$, $k_B T = 0.2\hbar\omega_P$, $N = 10^{10}$, $S = 1$, and the Rabi splitting is $\hbar\Omega = 5 \times 10^{-2} \hbar\omega_P$

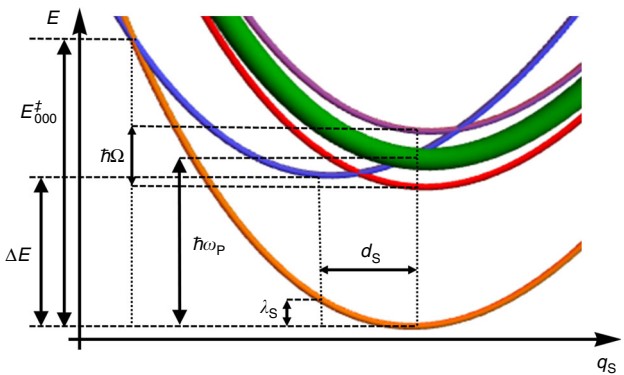

**Fig. 3** Potential energy surfaces under VSC along the slow coordinate. (Not to scale.) With respect to the reactant (blue), the vibrational ground state of the product (orange) is in the Marcus inverted regime; the manifold of states with one vibrational excitation (green, red, and purple) in the product is in the normal regime. Although the dark states (green) outnumber the lower (red) and upper (purple) polaritons, the small activation energy associated with the lower polariton channel might make it the preferred pathway for reactivity

absence of the latter, it becomes irrelevant for the cumulative kinetics, as we shall see next.

Up until now, we have shown that the rate coefficient depends on the number of molecules that take part in the VSC, which changes as the reaction progresses. To illustrate the cumulative effect on the kinetics, we numerically integrate the rate law

$$\frac{d\langle N_R \rangle}{dt} = -\langle k_{R\to P}^{VSC}(N_R) N_R \rangle \tag{18}$$

where $\langle \cdot \rangle$ indicates an average over the ensemble of reactive trajectories (see Supplementary Note 4). We show the behavior of $N_R(t) = M - N(t)$ for several detunings in Fig. 5. In writing Eq. (18), we have assumed that every electron transfer event is accompanied by a much faster thermalization of the products (largely into the global ground state in the products side) that allows us to ignore back-reactions. This assumption is well justified if we consider that, for systems with parameters close to our model molecule, the vibrational absorption linewidth is of the order of $0.01\hbar\omega_P$[12,19,40], which represents a timescale suitably

shorter than the reaction times estimated from the rate constant, $k_{R\to P} = 9.4 \times 10^{-6}\omega_P$, calculated with the same parameters. In Fig. 5, we can see that, for $\Delta \geq 0$, at early times the reactions proceed in the same way as in the bare case. However, after some molecules have been gathered in the product, the coupling is strong enough for the LP channel to open and dominate over the D ones. This effect is cumulative and the reaction endures a steady catalytic boost. Importantly, the maximum enhancement is observed for resonant conditions where the light–matter coupling is the most intense. On the other hand, with a slightly negative detuning, $\Delta = -0.02\omega_P$, the reaction is intensified in the early stages (as explained above) but is taken over by the dark states after a relatively short amount of time. Although this off-resonant effect might look appealing, it occurs at an early stage of the reaction when VSC is not technically operative, namely when the energetic separation between dark and polaritonic modes might be blurred by dissipative processes. These considerations are beyond the scope of the current article and will be systematically explored in future work. In conclusion, even though some off-resonant effects might be present at the rate coefficient level, the condition of resonance is essential to observe a significant cumulative acceleration of the reaction (i.e., change in reactant lifetime) with respect to the bare case.

Importantly, in the case where the high-frequency mode of the reactant molecules also couples to light, the system is under VSC before the reaction begins and the spectrum in the first excited manifold in the products remains invariant throughout the reaction. Therefore, the rate coefficient is a true rate constant evaluated at $N = M$, i.e., at the maximum coupling. We will present a detailed analysis of this problem elsewhere.

## Discussion

We have shown that VSC can result in catalysis of TA reactions. We have presented an MLJ model to study charge transfer processes under VSC (in passing, these results suggest a VSC alternative to enhance charge conduction, which has so far been only considered in the electronic strong coupling regime[37,46–49]). In this model, there is a range of molecular features where the shrinkage of the activation energy of the lower polariton channel can outcompete the rate associated with the massive number of dark-state channels. This model describes a mechanism suitable to be present in a wide variety of thermally activated non-adiabatic reactions, e.g., electron, proton, and methyl transfer, among

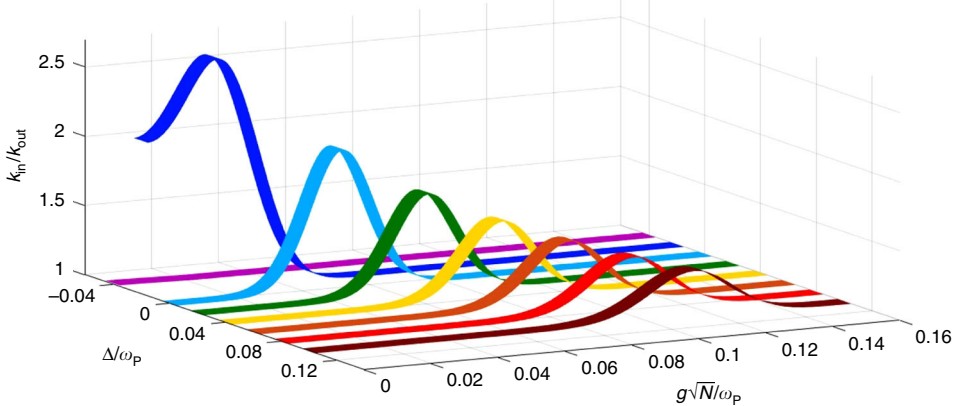

**Fig. 4** Ratio between rate coefficients inside and outside the cavity. The relation of $k_{in} = k_{R \to P}^{VSC}$ and $k_{out} = k_{R \to P}$ was calculated at several detunings $\Delta$. For these calculations $1 \leq N \leq 10^{11}$, $\omega_R = \omega_P$, $k_B T = 0.2\hbar\omega_P$, $S = 1$, $\hbar g = 1.6 \times 10^{-5}\hbar\omega_P$, and $E_{001}^{\ddagger} = 4.9\hbar\omega_P$. In agreement with Marcus theory, as the lower polariton mode decreases in energy (with increasing Rabi splitting), its corresponding activation energy falls and then rises, thus dominating the kinetics and becoming irrelevant, respectively. Notice that the trend of apparent enhancement at negative detunings eventually stops at low values of $|\Delta|$

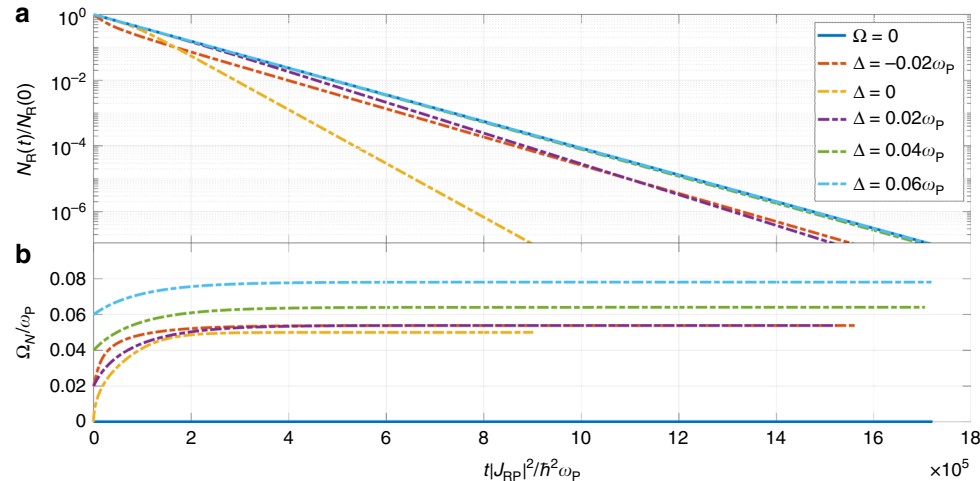

**Fig. 5** Evolution of reactant consumption. **a** Integrated rate law for the reaction outside and inside of the cavity at several detunings. The departure of the VSC enhanced kinetics with respect to the bare case becomes more significant at resonance. **b** Evolution of effective Rabi splittings as the reaction progresses. The effects on the kinetics are observed when the VSC regime ($\Omega_N > 0.01\omega_P$) is achieved. For these calculations, $M = N_R(0) = 10^7$, $\omega_R = \omega_P$, $k_B T = 0.2\hbar\omega_P$, and $E_{001}^{\ddagger} = 3.5\hbar\omega_P$

others. We have found a range of molecular parameters where the shrinkage of the activation energy of the lower polariton channel can outcompete the rate associated with the massive number of dark-state channels. We determined that these effects are most prominent under resonant conditions. This finding is relevant, as such is the behavior observed in experimental reactions performed under VSC. We must remark, however, that the latter are presumably vibrationally adiabatic reactions and the involvement of the present mechanism is not obvious (for a recent study on possibly important off-resonant Casimir–Polder effects, we refer the reader to ref. [50]). Although a thorough understanding of the reaction pathways involved in these observations is beyond the scope of this article, we believe that the tug-of-war between the activation energy reduction from few polariton channels against the numerical advantage of the dark states could be a ubiquitous mechanism of TA polariton chemistry under VSC, independently of whether it occurs with reactants or products. Even though there might be other subtle physical mechanisms underlying VSC TA reactions, we conclude with three important observations regarding the presently proposed catalytic mechanism. First, it

does not offer a reduction of reaction rate coefficients for a broad range of parameters; after all, if the polariton channels do not provide incentives for their utilization, the dark states will still be accessible, leading to virtually unaffected reaction rates as compared with the bare case. However, an experimental suppression of reactions by VSC under TA conditions (as in refs. [12,13]) could correspond, microscopically, to the polaritonic modification of elementary step rates in the network of reaction pathways that comprises the mechanism. Second, it is not evident whether the conclusions associated with this mechanism are relevant in photochemical processes where non-equilibrium initialization of polariton populations is allowed. Finally, it is important to emphasize that this VSC mechanism is not guaranteed to yield changes in TA reactivity, given that particular geometric molecular conditions need to be fulfilled. Regardless, it is remarkable that TA reactions under VSC can be modified at all given the entropic limitations imposed by the dark states. It is of much interest to the chemistry community to unravel the broader class of reactions and the VSC conditions for which this mechanism is operative; this will be part of our future work.

## Methods

**Numerical simulation.** To calculate the consumption of the reactant as the polaritonic ensemble grows, we performed a finite-difference numerical integration of Eq. (18). As the rate coefficient remains constant during a single-molecule event, we assume a mean-field ansatz

$$\langle k_{R \to P}^{VSC}(N_R) N_R \rangle(t) \simeq k_{R \to P}^{VSC}(\langle N_R \rangle(t)) \langle N_R \rangle(t),$$
$$\langle k_{R \to P}^{VSC}(N_R) N_R \rangle(t + \Delta t) \simeq k_{R \to P}^{VSC}(\langle N_R \rangle(t))(\langle N_R \rangle(t) - 1),$$
(19)

which enables the stepwise integration of Eq. (18) with limits $t \to t + \Delta t$ and $N_R \to N_R - 1$, yielding

$$\Delta t(N_R) = \frac{1}{k_{R \to P}^{VSC}(N_R)} \ln \frac{N_R}{N_R - 1}.$$
(20)

We verified that this mean-field method gives numerically consistent results with the stochastic simulation algorithm (see Supplementary Note 4 and Supplementary Table 1)[51], in agreement with recent studies of mean-field solutions to polariton problems in the ensemble regime[52]. The rate coefficient $k_{R \to P}^{VSC}(N_R)$ at each step is calculated from Eq. (11) truncating the sum up to $\nu_+ = \nu_- = \nu_D = 2$; terms beyond these excitations do not contribute appreciably given their huge activation energies resulting from the chosen parameters. The Franck–Condon and exponential factors are calculated, respectively, from Eq. (12) and Eq. (13) by setting $\omega_R = \omega_P$.

## Data availability

Data sharing not applicable to this article as no datasets were generated or analysed during the current study.

## Code availability

The source code for the simulation of the modified kinetics is avialable from the corresponding author upon request.

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

## Acknowledgements

J.A.C.G.A. thanks Matthew Du and Luis Martínez-Martínez for useful discussions. J.A.C. G.A. acknowleges initial support of the UC-MEXUS-CONACYT graduate scholarship ref. 235273/472318. J.A.C.G.A. and R.F.R. were sponsored by the AFOSR award FA9550-18-1-0289. J.Y.Z. was funded with the NSF EAGER Award CHE 1836599. All the authors thank Blake Simpkins at NRL and Jonathan Keeling at University of St. Andrews for their insightful comments about the manuscript.

## Author contributions

J.A.C.G.A. carried out the theoretical setup of the electron transfer reaction in terms of polariton and dark modes. J.A.C.G.A. and R.F.R. calculated the Franck–Condon factors and developed the Marcus–Levich–Jortner model. J.Y.Z. provided guidance on the formulation of the model and the interpretation of the results. All the authors contributed equally to the manuscript.

## Competing Interests

The authors declare no competing interests.
