## [Peer Review File · Nature Communications]

Reviewers' comments:

Reviewer #1 (Remarks to the Author):

The manuscript by Campos-Gonzalez-Angulo et al. reports theoretical calculations of how vibrational strong coupling between a cavity and the product of a chemical reaction can enhance reaction rate. The manuscript considers carefully the competition between the fact that polaritonic states make up only a small fraction of possible final vibrational states, and the energetic gain associated with using polaritonic states. The authors show that this does allow an enhancement to occur under reasonable conditions, despite the small fraction of final vibrational states which are polaritonic. This makes an important contribution to the theoretical understanding of this highly topical area, and I therefore recommend it for publication in Nature Communications.

There are a few minor issues the authors may wish to address before publication:

* Above Eq. 1 the authors introduce donor and acceptor states labeled by R and P. Clearly the labels R and P are for reactant and product, however that language is not used here. I think it would help the reader (particularly those from a physics background) if this were spelled out when these labels are first introduced, otherwise it is unclear why donor should be labeled R and acceptor labeled P.

* In explaining the terms in Eq. 2, the appearance of the squeezing operator is perhaps not entirely obvious. Its origin is clearly explained in the supplementary information. I would suggest an explicit reference here to say "See supplementary information for the origin of this term."

* At the end of the paragraph mentioned above, there is a reference to polaritonic effects in electron transfer processes, which cites Ref. 26, Herrera and Spano, "Dark vibronic polaritons and the spectroscopy of organic microcavities". I wonder if that is really the paper the authors meant to cite, or whether they meant Herrera and Spano, "Cavity-Controlled Chemistry in Molecular Ensembles" Phys. Rev. Lett. 2016, 116, 238301. [NB, there is a later reference to Ref. 26 which seems correct; it is only this citation where I think another paper was intended.]

* On page 3, the notation $D^{\{(N-1)\}}_{\{N-1\}}$ is used three times, on the right of Eq. 7, in the paragraph below it, and above Eq. 11. In all three places I believe the authors meant $D^{\{(N-1)\}}_{\{N\}}$. i.e. this is the N-1 th dark state in the manifold where N product molecules exist.

* In Eq. 15, I believe ν_P should be ν_D in the last term.

* In discussing Fig. 5, the authors state that while the reaction starts faster for negative cavity detuning, the resonant case shows the greatest cumulative effect on reaction rate. This is clearly true in the way this has been defined here, but it is not obvious this is the most relevant practical consideration. From Fig. 5, if one were interested in attaining 90% yield as fast as possible, it seems the negative detuning would actually be best. It is only when one starts demanding 99% yield that the resonant case becomes optimal. i.e., for practical purposes of attaining a reasonably large yield fast, it may be better to optimize the early stage of the reaction, and not worry about the time to get from 99% to 99.9999% (corresponding to 10^{-6} fraction of reactant shown on the range of the graph).

Reviewer #2 (Remarks to the Author):

The manuscript presents a theoretical model for non-adiabatic electron transfer (ET) based on the Marcus-Levich-Jortner theory where a high frequency intermolecular vibration is strongly coupled to an electromagnetic mode of a cavity resonator. The coupling between the mode and the vibration is treated within the Jaynes-Cummings model. The vibrational strong coupling (VSC) between the N th identical vibrations and the cavity mode leads to the formation of collective hybridized light-matters states (polaritons), where $N-1$ states are so-called dark states which energy levels are the same as for the uncoupled vibrations and they do not “mix” with the mode and two bright states (upper and lower polaritons). The authors suggest that under specific conditions the ET rate can be substantially enhanced when the reaction pathway via the excitation of the lower polariton which has a smaller activation energy becomes dominant and provide some calculations in support.

If substantiated, this finding offers a new possible mechanism for cavity strong vibrational coupling effects on chemical kinetics, thereby of great interest to the community. However, the compact format of a nature communication article makes the presentation, which is theoretically involved, not very clear. Without going into this issue there are key points that need to be addressed before the article is further considered. First, the picture presented by the authored is based on polariton formation due to coupling of the group vibrational modes associated with the product molecules to the cavity modes. I find this picture hard to understand. In standard theories of electron transfer there are no “product modes” and “reactant modes”. There are modes that couple to the electronic transition. One could perhaps construct a theory based on the difference between ω_R and ω_P but it does not appear that the authors are doing this, and rightly so – such frequency differences are

rather small and a very careful analysis will be needed to draw conclusions based on such differences. Now, if the cavity mode is coupled equally to all modes, it affects the multidimensional potential surfaces of the donor and acceptor state in the same way. It cannot therefore affect the reorganization energy and will not have an effect on the electron transfer. If one build a picture in which ω_R and ω_P are so different that coupling to the cavity changes upon electron transfer, it still appears that for a single reaction event $A \rightarrow B$ where one molecule goes from a donor state to an acceptor state, the difference between the potential surfaces $V(B)-V(A)$ (now $V(A)$ and $V(B)$ include coupling to cavity mode and this coupling is linear in the molecular coordinates) is given by just the molecule that made the transition, so it is hard to see how the effect can be of order N .

It is possible that I miss something in the authors' arguments, and it will be good to see their reply. At present, I cannot recommend this paper for publication, and it is possible that even if the authors successfully defend their picture, it will be better for this paper to eventually be published as a regular size paper where arguments can be made clear.

Reviewer #3 (Remarks to the Author):

The authors are trying to develop theoretical model for explaining chemistry under vibrational strong coupling (VSC). In this latest paper the starting point is the Marcus-Levich-Jortner electron transfer theory which they apply to VSC and they get some very interesting results, especially in regards to relative role of dark states, the $n-1$ collective states that are formed when one optical mode couples to many molecules. While this paper and the theoretical part is perfectly suitable for publication in Nature communication, it requires major revision on a number of points in regards to the relevance to published papers:

- 1) The published experimental papers (ref 12-15) on chemistry under VSC are not simple electron transfer reactions but involve complicated transition states with bonds breaking and forming. The authors have to change their introduction and discussion to reflect this so as to not confuse readers even if any well-trained chemist will know this. As written, it only diminishes the value of the paper. The paper makes prediction which are good enough by themselves and when proper electron transfer reactions are studied it will be very interesting to compare with this work.
- 2) It is postulated that only a "tiny fractions of the molecules will be allocated to polariton modes" in the second paragraph. If this was the case, then slower reactions would never be observed under VSC contrary to observations. The authors come back to this point in the discussion, last column, and try to get around this logic inconsistency with the proposition that there are unknown competing side reactions. This is presumptuous on two levels: it assumes that the simple

electron transfer model applies to non-electron transfer reactions (point 1), secondly that the experimentalists can't detect such a side-reactions with their chemical analysis techniques.

3) The authors add to the confusion on dark-state reservoir by stating that the "overwhelming majority (of the population is) residing in the dark-state reservoir" as if at room-temperature the collective modes generated by VSC are populated. The occupation factor is tiny for all these states as a simple Boltzmann calculation shows. The great majority are in the $v=0$ state.

The authors have to be more precise to add to the existing confusion as discussed in ref. 20.

4) Minor: add Semenov and Nitzan reference on electron transfer in confined electromagnetic fields JPC 150, 174122 (2019); Run spell check... e.g. p3 column 1: "diabtic"

Responses to the referees

In the first place, we would like to thank the referees for their detailed and thorough reviews of our manuscript. We are glad that they found that our work “makes an important contribution to the theoretical understanding of this highly topical area,” (Referee 1) with the potential of being “of great interest to the community” (Referee 2), and that it “is perfectly suitable for publication in Nature communication” (Referee 3). We believe that the new version of the manuscript has been substantially improved by virtue of the referees’ comments. In the following, we discuss the comments and suggestions raised by the referees (in blue) and include the modifications to the manuscript (in red).

Reviewer 1

1. Above Eq. 1 the authors introduce donor and acceptor states labeled by R and P. Clearly the labels R and P are for reactant and product, however that language is not used here. I think it would help the reader (particularly those from a physics background) if this were spelled out when these labels are first introduced, otherwise it is unclear why donor should be labeled R and acceptor labeled P.

We thank the referee for raising this concern. Indeed, our model describes situations that are more general than charge transfer from a donor to an acceptor; instead, it describes a generic nonadiabatic thermally activated process between two electronic states of different chemical character. According to this understanding, we have changed the language from donor/acceptor to reactant/product throughout the text.

2. In explaining the terms in Eq. 2, the appearance of the squeezing operator is perhaps not entirely obvious. Its origin is clearly explained in the supplementary information. I would suggest an explicit reference here to say "See supplementary information for the origin of this term."

Done.

3. At the end of the paragraph mentioned above, there is a reference to polaritonic effects in electron transfer processes, which cites Ref. 26, Herrera and Spano, "Dark vibronic polaritons and the spectroscopy of organic microcavities". I wonder if that is really the paper the authors meant to cite, or whether they meant Herrera and Spano, "Cavity-Controlled Chemistry in Molecular Ensembles" Phys. Rev. Lett. 2016, 116, 238301. [NB, there is a later reference to Ref. 26 which seems correct; it is only this citation where I think another paper was intended.]

Indeed, the reference corresponds to Phys. Rev. Lett. 2016, 116, 238301. Changed.

4. On page 3, the notation $D^{\{(N-1)\}}_{\{N-1\}}$ is used three times, on the right of Eq. 7, in the paragraph below it, and above Eq. 11. In all three places I believe the authors meant $D^{\{(N-1)\}}_{\{N\}}$. i.e. this is the N-1 th dark state in the manifold where N product molecules exist.

We thank the referee for noticing this. Furthermore, we realized that this notation could be somewhat confusing; therefore, we decided to change it in the definition of the dark mode basis (see Eqs. 3 and 6) such that the subscript j in $D_j^{(k)}$ still indicates the number of product molecules while the superscript k now indicates the molecule in which that dark mode is mainly localized.

5. In Eq. 15, I believe ν_P should be ν_D in the last term.

Thanks, we have corrected this typo.

6. In discussing Fig. 5, the authors state that while the reaction starts faster for negative cavity detuning, the resonant case shows the greatest cumulative effect on reaction rate. This is clearly true in the way this has been defined here, but it is not obvious this is the most relevant practical consideration. From Fig. 5, if one were interested in attaining 90% yield as fast as possible, it seems the negative detuning would actually be best. It is only when one starts demanding 99% yield that the resonant case becomes optimal. i.e., for practical purposes of attaining a reasonably large yield fast, it may be better to optimize the early stage of the reaction, and not worry about the time to get from 99% to 99.9999% (corresponding to 10^{-6} fraction of reactant shown on the range of the graph).

While we agree with the keen insights made by the referee, we do not want to emphasize this effect because we worry that at early stages of the reaction, the light-matter coupling is weak and finite linewidth effects might blur the energetic separation between the polariton and the dark modes, and thus erase this effect. We have now included this hypothesis in our discussion (p. 6, left column): “Although this off-resonant effect might look appealing, it occurs at an early stage of the reaction when VSC is not technically operative, namely when dissipative processes might blur the energetic separation between dark and polaritonic modes. These considerations are beyond the scope of the current article and will be systematically explored in future work.” We would like to postpone a more systematic discussion of these off-resonant effects in future work.

Reviewer 2

The picture presented by the authors is based on polariton formation due to coupling of the group vibrational modes associated with the product molecules to the cavity modes. I find this picture hard to understand. In standard theories of electron transfer there are no “product modes” and “reactant modes”. There are modes that couple to the electronic transition. One could perhaps construct a theory based on the difference between ω_R and ω_P but it does not appear that the authors are doing this, and rightly so – such frequency differences are rather small and a very careful analysis will be needed to draw conclusions based on such differences.

We thank the referee for making us aware that this could be a confusing language in our work. Our model is actually in agreement with your statement: it describes vibrational modes that couple to the electronic transition.

The equivalent terminology we have used is based on the diabatic representation where each electronic state consists of a harmonic oscillator along each vibrational mode. The multidimensional potential energy surfaces for the product state is a displaced-rotated version of the reactant state. By reactant and product modes, we simply mean the harmonic oscillator potentials for each electronic state.

In our model, the electronic transition couples to the modes not only through displacement operators but also through Duschinsky rotations that mix the photon and the high-frequency molecular mode. Our model assumes that there is a switch in IR activity upon electron transfer: the high-frequency molecular mode is IR inactive in the reactant side and IR active in the product side. Thus, the polariton modes are the eigenmodes in the product state. A physical origin of this switch in IR activity occurs when dramatic changes in dipole moment follow from charge transfer, such as in molecular actuators like cyclotetrameratrylene (*Org. Biomol. Chem.*, 2018, **16**, 5712) and tetramethoxydibenzobicyclo[4.4.1.]undecane (*J. Phys. Chem. B* 2010, **114**, 14592–14595). We have included these references in the manuscript to clarify any misunderstandings derived from this behavior (p. 2, right column):

“This constraint implies a drastic change in molecular geometry upon charge transfer so that the vibrational transition dipole moment goes from negligible to perceptible. This rather unusual behavior can be observed in molecular actuators.[24, 25]”

On the other hand, as the referee accurately points out, the quantitative analysis of the consequences of the model is performed under the condition that $\omega_R = \omega_P$; however, our formalism is general enough to consider a shift in frequency. The latter is achieved by the inclusion of a squeezing operator in the coupling between the modes and the electronic transition.

Finally, it is worth noting that we have recently considered the case where the IR activity of the high-frequency mode is the same in both reactant and product side, and we have found that, under those circumstances, the inequality in Eq. 16 remains the same, so the conclusions do not change under those circumstances, except for the fact that, under those circumstances, the energy barrier to the LP mode does not change as a function of time. There are several technical differences with respect to the present theory; so, those findings will be published soon elsewhere.

Now, if the cavity mode is coupled equally to all modes, it affects the multidimensional potential surfaces of the donor and acceptor state in the same way. It cannot therefore affect the reorganization energy and will not have an effect on the electron transfer.

This is an interesting observation. This is indeed the case when all the vibrational modes are treated classically (high-temperature limit and short time approximation to the rate theory) and the modes couple to the electronic excitation only through displacements for both inner and outer sphere reorganization energies (standard Marcus theory). Having said that, we are now aware that the consideration of frequency shifts (going from reactant to product) through the inclusion of squeezing operators leads to changes to the reorganization energies upon coupling to the cavity mode, even within the classical treatment. We include this exploration in our work in progress.

Regardless, it is well known that quantization of high-frequency modes significantly affects charge transfer rates (MLJ formalism). In the conditions where the MLJ formalism is more accurate than the traditional Marcus theory, the charge transfer can happen from the potential energy surface of the reactant ground state to those in which the product includes vibrational excitations in its high-frequency modes. The strong coupling between these high-frequency modes and the electromagnetic field is the one we are exploiting for our polaritonic mechanism.

If one builds a picture in which ω_R and ω_P are so different that coupling to the cavity changes upon electron transfer, it still appears that for a single reaction event $A \rightarrow B$ where one molecule goes from a donor state to an acceptor state, the difference between the potential surfaces $V(B)-V(A)$ (now $V(A)$ and $V(B)$ include coupling to cavity mode and this coupling is linear in the molecular coordinates) is given by just the molecule that made the transition, so it is hard to see how the effect can be of order N .

While we agree that each reaction is a single molecule event, the mechanism that we describe is cumulative. In the coupling scheme that we described in our paper, the transition occurs from the $N + 1$ -dimensional ground state PES (with 1 reactant, $N - 1$ products, and the cavity mode) to the $N + 1$ -dimensional PES (with N products and the cavity mode). The accumulation of molecules in the product configuration is what gives rise to the collective strong coupling. Hence, the effect of VSC is cumulative and manifests in an observable extent only after a substantial number of product molecules have been formed, i.e., at intermediate stages of the reaction.

At present, I cannot recommend this paper for publication, and it is possible that even if the authors successfully defend their picture, it will be better for this paper to eventually be published as a regular size paper where arguments can be made clear.

We hope the referee finds our responses satisfactory to defend our picture. Moreover, we consider that the communications format is appropriate for our work on the basis that:

- the arguments, discussions and added clarifications stand on their own, and the main message is transmitted without requiring further elaboration,
- the results and their implications are novel and of broad interest for the scientific community (a fact acknowledged by all the reviewers).

While we recognize that there are many aspects of this topic open for exploration, we think that each of these questions merits a full paper on its own.

Reviewer 3

- 1) The published experimental papers (ref 12-15) on chemistry under VSC are not simple electron transfer reactions but involve complicated transition states with bonds breaking and forming. The authors have to change their introduction and discussion to reflect this so as to not confuse readers even if any well-trained chemist will know this. As written, it only diminishes the value of the paper. The paper makes prediction which are good enough by themselves and when proper electron transfer reactions are studied it will be very interesting to compare with this work.

We thank the referee for this insight and their comments about the merit of our results. We have modified the introduction and the discussion to increase the distance between our work and those experiments and be more explicit about their differences (p. 1, left column):

“This picture could potentially change as a consequence of ultrastrong coupling effects; however, these effects should not be significant for modest Rabi splittings as those observed in the experiments.”

(p. 6 ,right column):

“This model describes a mechanism suitable to be present in a wide variety of thermally activated nonadiabatic reactions, e.g., electron, proton and methyl transfer, among others. We have found a range of molecular features where the shrinkage of the activation energy of the lower polariton channel can outcompete the rate associated with the massive number of dark-state channels. We determined that these effects are most prominent under resonant conditions. This finding is relevant since such is the behavior observed in experimentally in reactions performed under VSC. We must remark, however, that these are vibrationally adiabatic reactions and the involvement of the present mechanism is not obvious (for a recent study on possibly important off-resonant Casimir-Polder effects, we refer the reader to [50]).”.

However, the changes are not substantial since we consider that the fact that it highlights the roles of resonance and concentration is one of the most significant achievements of our model since this is consistent with the trends observed in the experiments.

- 2) It is postulated that only a “tiny fractions of the molecules will be allocated to polariton modes” in the second paragraph. If this was the case, then slower reactions would never be observed under VSC contrary to observations. The authors come back to this point in the discussion, last column, and try to get around this logic inconsistency with the proposition that there are unknown competing side reactions. This is presumptuous on two levels: it assumes that the simple electron transfer model applies to non-electron transfer reactions (point 1), secondly that the experimentalists can't detect such a side-reactions with their chemical analysis techniques.

We appreciate that the referee has made us aware of this interpretation of our discussion. First, we stand by our claim about populations at thermal equilibrium, which is actually in agreement with the referee's next comment (point 3) concerning Boltzmann populations. As our results indicate, the role of the polariton modes may be relevant regardless of the thermal distribution. The polariton excitations instead serve as channels with low-activation barriers through which reactants become products. Second, while we agree with the referee that our model does not address so-called “vibrationally adiabatic” reactions that are described by transition state theory, we claim that it describes a wide class of thermally activated “nonadiabatic reactions”. In these processes, the reactant needs to accumulate enough energy to reach a crossing between two potentials of different chemical character, upon which an electrostatic interaction

transfers population from reactant to product; these are well described by Marcus-type approaches. The modification we implemented on the discussion addresses this concern.

On the other hand, we are very much aware that current experimental reports have elucidated the activation of competing reaction pathways through chemical kinetics techniques (e.g., SN1 vs. SN2, or the Si-C vs. Si-O bond breaking). Following the referee's recommendations, and to avoid inconsistencies with the reported experiments, we have rephrased our claim in the manuscript (p 6, right column):

“However, an experimental suppression of reactions by VSC under TA conditions (as in [12, 13]) could correspond, microscopically, to the polaritonic modification of elementary step rates in the network of reaction pathways that comprises the mechanism.”

3) The authors add to the confusion on dark-state reservoir by stating that the “overwhelming majority (of the population is) residing in the dark-state reservoir” as if at room-temperature the collective modes generated by VSC are populated. The occupation factor is tiny for all these states as a simple Boltzmann calculation shows. The great majority are in the $v=0$ state. The authors have to be more precise to add to the existing confusion as discussed in ref. 20.

We agree with the referee's point, and we have clarified this source of confusion. We have rewritten the beginning of that paragraph to indicate that we are comparing only populations of excited states (p. 1 right column):

“From the population of vibrationally excited states at thermal equilibrium, a tiny fraction would be allocated to the polariton modes, with the overwhelming majority residing in the dark-state reservoir [17–20].”

4) Minor: add Semenov and Nitzan reference on electron transfer in confined electromagnetic fields JPC 150, 174122 (2019); Run spell check... e.g. p3 column 1: “diabtic”

We thank the referee for this suggestion. We have updated our references and made the appropriate corrections.

Reviewers' comments:

Reviewer #1 (Remarks to the Author):

The revised manuscript by Campos-Gonzalez-Angulo et al. satisfactorily makes all the minor corrections I had noted previously. As such, I believe the manuscript should be published once the authors have made the few extra minor corrections noted below,

After reading the comments of the other referees, I feel I should explain further my enthusiasm for this manuscript. I believe this manuscript makes a significant contribution in a number of ways:

Firstly, it takes seriously the nature of dark polaritonic states, and the counting of such states. This is important as this manuscript explains, because in general there are many more states that are unshifted by strong matter light coupling than those which are. If such physics is not seriously addressed, then it is not possible to really test whether a proposed mechanism actually competes with the background rates due to dark states. While it is clear from experimental results that strong vibrational coupling does have an effect of reaction rates, that does still leave open questions of mechanism, and any proposed mechanism must be compatible with the physical reality of the dark states.

Secondly, the manuscript is significant in combining a careful analysis of the nature of polaritonic states with a more thorough consideration of the mechanism of thermally assisted reactions, by considering the Marcus-Levich-Jortner formalism. This goes beyond the toy models used in many theoretical papers in this field. As shown in the manuscript, this approach also has significant effects on the reaction rate. i.e., it introduces the idea of strong coupling inducing a crossover between the Marcus inverted and normal regimes. I feel this is also a significant contribution to the theoretical understanding of such problems.

Regarding one comment of another referee, I note that it might be helpful if the statement made by the authors on page one, about the fraction of bright and dark states, was made more quantitative. i.e., since this fraction depends on the ratio of the molecule number N and the Boltzmann factor $\exp(-\beta \cdot \Omega)$, there can be no absolute statement that the fraction is always small. i.e., at low temperatures, the polariton state will dominate. By restating this point as behavior for typical couplings at room temperature, one can estimate the Boltzmann factor, and compare it to typical values of N . This will make the authors statement more precise.

In re-reading the manuscript I noticed a few other minor points the authors should fix before publication:

* Page 1, "we find a parameters range" -> "we find a parameter range"

* Page 4, There are two references to Fig. S2 which I believe are meant to be references to Fig. 3. (Fig. S2 is an enlargement of Fig.3, but it is better to refer to the figure in the paper rather than the supplement.)

* Page 6, new text, "range of molecular features" should probably be "range of molecular parameters".

* Supplement, Eq. S14 should have ν_- as exponent in the second term.

Reviewer #2 (Remarks to the Author):

I am sorry that I can still not recommend this manuscript for publication. From my perspective the authors did not respond adequately to my concerns.

The authors consider a Harmonic model in which molecular electron transfer is associated with a change in the molecular (assume harmonic) potential surface. They use a general model in which the differences between reactant and product surfaces may include origin shift, frequency changes and rotations. All modes can couple to one or more cavity modes.

The issue is understanding the origin of the effect predicted by the authors. The paper starts by using a language by which the modes of product (but not the reactant) molecules are coupled through the radiation field. In my original response I expressed concern about the following issues: (a) I argued that in the classical (Marcus) the cooperative effect predicted by the authors will not exist. (b) I pointed out that the model does not comprise reactant modes and product modes, but reactant and product harmonic surfaces are sensitive to the electronic state. I also noted that, while the situation where the potential surfaces change substantially upon electron transfer so that strong coupling to a cavity mode is manifested only in the product surface is interesting and merits analysis, it appears that this is not the focus of the study since the author claim that their effect exists also in the simpler shifted-only surfaces model. Indeed the latter model is the focus of their numerical studies.

The authors appear to agree with my point A, and now put their focus on the high frequency modes that should be treated quantum mechanically. They also accommodate their *language* to address my concern. However, the *substance* of their analysis remains the same. So here is what I am concerned about: To understand the origin of their effect I can apply the model in its simplest limit. Since they now focus on the high frequency mode I will consider only such modes and still stick to the shifted-origin-only model that can be solved analytically. In this limit the vibronic coupling is essentially a product of an electronic matrix element and a Frank-Condon (FC) factor. The question is whether, for the process where a single molecule changes its electronic state, the latter is modified due to coupling to the cavity mode. I believe that at least in this limit I can show that the answer is negative. I cannot say whether there is a more interesting behavior in situations that involve frequency changes and rotations. The authors may find interesting behaviors in such cases, but since in the current analysis they predict an important effect where I think that none can exist, I cannot recommend this publication.

Reviewer #3 (Remarks to the Author):

The authors have answered my queries satisfactorily and the manuscript now deserves acceptance. Please update reference 15 as it is now available online in *Angewandte Chemie*:
<https://doi.org/10.1002/anie.201905407>

Rebuttal letter

First and foremost, we would like to express our appreciation for the time and dedication that each of the reviewers invested in our work, as evidenced by their thorough and insightful comments. Regarding Reviewer #2, whose unfavorable assessment motivates this letter, we want to acknowledge that their comments expose a subtle yet relevant source of confusion in prior versions of the manuscript and have therefore prompted us to improve the clarity about some nuances in our discussion. In what follows, we address the concern raised by Reviewer #2 and show that the substance of our work, contrary to their suggestion (and in agreement with Referees #1 and #3) *does not need to be revised*.

Reviewer #2 starts their report with the following sentences:

"I am sorry that I can still not recommend this manuscript for publication. From my perspective the authors did not respond adequately to my concerns."

We respectfully disagree with the reviewer's perspective. As we will show going forward, we successfully answered their questions and clarified several misunderstandings between the previous and current rounds.

The reviewer states:

*"I expressed concern about the following issues: (a) I argued that in the classical (Marcus) the cooperative effect predicted by the authors will not exist... The authors appear to agree with my point A, and now put their focus on the high frequency modes that should be treated quantum mechanically. They also accommodate their *language* to address my concern. However, the *substance* of their analysis remains the same."*

Seeing that Reviewer #2 now appreciates that our results rely on a model of electron transfer that goes beyond the classical (Marcus) one, we conclude that we have clarified the reviewer's interpretation of the first version of the manuscript. Indeed, we agree with point (a), but we have made it very clear that this is not the situation that we are exploring. Since our formalism has consistently been based on the quantum version of Marcus theory (Marcus-Levich-Jortner, MLJ), in which the quantization of the high-frequency modes plays an essential role, **the focus of our work has always explicitly been the high-frequency modes**. However, given the confusion that Reviewer #2 manifested in the first stage of review, we amended our manuscript ("accommodated our language") in the submitted revision.

The reviewer also says:

"I also noted that, while the situation where the potential surfaces change substantially upon electron transfer so that strong coupling to a cavity mode is manifested only in the product surface is interesting and merits analysis, it appears that this is not the focus of the study since the author claim that their effect exists also in the simpler shifted-only surfaces model. Indeed, the latter model is the focus of their numerical studies."

While most of the description provided by the reviewer is accurate, which incidentally shows an improvement of their understanding resulting from our adequate response, there is a noteworthy aspect of it that needs clarification. Our example calculation assumes for simplicity that the bare molecular frequencies of the vibrations do not change when going from reactant to product (so, in fact, it corresponds to a "shifted-only surfaces model" for the bare molecules outside of the cavity). However, the transition dipole moment of the high-frequency mode is assumed to change going from reactant to product, **such that only the product affords strong light-matter coupling with the cavity**. This assumption is quite natural, as changes in the permanent dipole moment of the electronic state going from reactant to product can easily arise due to rearrangement of charges. In fact, we have explained this feature of our model since the first version of the article, and we have emphasized it even more in the first round of reviews

On the other hand, the main criticism of Reviewer #2 is contained in the following sentences to which we will present a detailed response:

“(b) I pointed out that the model does not comprise reactant modes and product modes, but reactant and product harmonic surfaces are sensitive to the electronic state... To understand the origin of their effect I can apply the model in its simplest limit. Since they now focus on the high frequency mode, I will consider only such modes and still stick to the shifted-origin-only model that can be solved analytically.”

We note that, although being quite comprehensive regarding the basics of electron transfer, the reviewer's criticism fails to capture the richness in our formalism. To give a proper response to the point raised by the reviewer in these sentences, we would first like to reiterate that the spotlight of our formalism has always been on the high-frequency vibrational quantum mode; there has not been any shift in our focus as the reviewer seems to suggest. We completely agree with point (b). **We never implied that there were reactant and product vibrational modes; in fact, in Eqs. (8 - 10) there is a single set of vibrations (\hat{a}_0 , $\hat{a}_{B(N-1)}$ and \hat{a}_N) that are rotated, displaced and squeezed when going from reactants to products.**

Furthermore, we emphasize that the sensitivity of the harmonic surfaces to the electronic state, to which Reviewer #2 refers in their comment, can explain the selectivity of the electronic states to the coupling with the cavity mode by considering that **the electronic transition inside the cavity necessarily implies a Duschinsky rotation involving the high-frequency reactive mode and the electromagnetic one.** Therefore, the “shifted-origin-only model” is accurate exclusively for bare molecules.

We emphasize that in the MLJ model, the transition from reactants to products involves coupling to both low and high-frequency modes, in agreement with the understanding of Reviewer #2. As it is made explicit in Eq. (2) of the manuscript, the surfaces related to the classical coordinates only experience displacement upon electronic transition. In contrast, the surfaces related to the quantum coordinates experience, upon electronic transition, displacement, and a cavity-dependent Duschinsky rotation, with the option to also include a frequency shift (we opted for the simplest case with constant frequency for our numerical arguments). Consequently, by virtue of this cavity-dependent Duschinsky rotation, a “shifted-origin-only model” is inapplicable for our polariton problem.

“In this limit the vibronic coupling is essentially a product of an electronic matrix element and a Frank-Condon (FC) factor. The question is whether, for the process where a single molecule changes its electronic state, the latter is modified due to coupling to the cavity mode.”

While it is true that the vibronic coupling is described by the product of a matrix element with an FC factor, it seems that the referee invokes a picture of chemical reactions relying on standard treatments in which every molecular transformation is an event independent from the others. **This depiction ignores the collective nature of the light-matter coupling that makes an appearance in the process as described in our manuscript.** The non-trivial and nonlinear dependence of the rate law on the number of product molecules that we bring to the limelight in the numerical section of the paper indicates a clear departure from the traditional picture.

“I believe that at least in this limit I can show that the answer is negative.”

The referee does not present any technical objection to our derivations in the paper; moreover, their only counterargument is a speculative proof that is absent in the review. In an effort to make things even more transparent, we have included a new section in the Supplementary Information accompanying the manuscript (and that we attach to the end of this letter) that specifically addresses the proof proposed by the reviewer. In it, **we show that the coupling to the cavity mode indeed modifies the FC factor** to which they refer, thus definitely refuting their statement.

“The authors may find interesting behaviors in such cases, but since in the current analysis they predict an important effect where I think that none can exist, I cannot recommend this publication.”

Again, we emphasize the subjective character of the criticism by Reviewer #2. To be clear, we concede that opinions play a role in academic discussion. However, we find it unsettling that, regardless of admitting the idiosyncratic component in their assessment, the reviewer opted to categorically withhold their support instead of keeping inquiring about their concerns.

On a different note, we must remark that our model is generic; hence, it allows a straightforward generalization to the case accounting for additional Duschinsky rotations that might be present in the bare case. The effect of such transformations could both enhance or decrease the effect described in the manuscript, but since this should be molecule-dependent, we plan to address this issue in a separate detailed paper where we work on a specific molecule. At this point, our focus is on showing to the community that there is a potential mechanism to enhance chemical reactions through an auto-catalytic strong-coupling mediated process.

We summarize by reiterating that **the novel idea in our paper is that the changes induced by infrared strong coupling on the kinetics of charge transfer can be large and thus significantly affect the rate of this process. We do not only prove the latter but also provide specific conditions where we expect the discussed effects to be more prominent.**

In the present discussion, we have shown that the substance of our work in its current form is enough to provide a definitive answer to the reviewer's question. Therefore, we assess that there should be no more, in words of the editor, "important concerns regarding the validity of the proposed model," nor "inconsistencies when the model is applied in the limit of considering only the high-frequency modes."

Given the above, in addition to the approval of Reviewer #3 and the enthusiastic endorsement of Reviewer #1, we consider that the work is in an appropriate stage for its publication in Nature Communications.

Yours sincerely,

Joel Yuen-Zhou,

Assistant Professor

Department of Chemistry and Biochemistry,

University of California San Diego

on behalf of all co-authors.

Initial and final many-body vibronic states

The rate to calculate corresponds to the stoichiometric process

where N is the number of molecules in the product electronic state P , and $M - N$ is the number of molecules in the reactant electronic state R , such that M is the total number of molecules in the reaction vessel. Assigning labels to each molecule, without loss of generality, the transformation of the $N + 1$ -th molecule can be written in the form

which reduces to

The adiabatic coupling $\hat{J} = J_{RP} \sum_{i=1}^M (|R_i\rangle\langle P_i| + |P_i\rangle\langle R_i|)$ rules the charge transfer; then, the matrix element that describes the process of our focus is

$$\langle M - N, N | \hat{J} | M - N - 1, N + 1 \rangle = J_{RP} \langle M - N, N | R_{N+1} \rangle \langle P_{N+1} | M - N - 1, N + 1 \rangle,$$

with many-body vibronic states given by

$$|X, Y\rangle = |P_1 P_2 \dots P_{Y-1} P_Y R_{Y+1} R_{Y+2} \dots R_{X+Y-1} R_{X+Y}\rangle \otimes |\Phi_X^Y\rangle,$$

where $|\Phi_X^Y\rangle$ is an eigenfunction of a vibrational Hamiltonian of the form

$$\begin{aligned} \hat{H}_{X,Y} &= \hat{H}_{\text{ph}} + \sum_{i=1}^Y (\hat{H}_P^{(i)} + \hat{\mathcal{V}}_{\text{int}}^{(i)}) + \sum_{j=Y+1}^{X+Y} \hat{H}_R^{(j)} \\ &= \hat{H}_{+(Y)} + \hat{H}_{-(Y)} + \sum_{k=1}^{Y-1} \hat{H}_{D(Y)}^{(k)} + \sum_{i=1}^Y \hat{H}_S(\hat{\mathbf{q}}_S^{(i)}) + Y\Delta E + \sum_{j=Y+1}^{X+Y} \hat{H}_R^{(j)}. \end{aligned}$$

In the equation above, we have used the notation introduced in Eq. (2) of the main body, while $\hat{H}_{\pm(Y)} = \hbar\omega_{\pm} (\hat{a}_{\pm(Y)}^{\dagger} \hat{a}_{\pm(Y)} + \frac{1}{2})$ and $\hat{H}_{D(Y)}^{(k)} = \hbar\omega_p (\hat{a}_{D(Y)}^{\dagger(k)} \hat{a}_{D(Y)}^{(k)} + \frac{1}{2})$ are the Hamiltonians of the upper/lower and k -th dark modes, respectively, all with creation and annihilation operators as defined in Eq. (3) of the main body. Therefore, the matrix element corresponding to the transition becomes

$$\langle M - N, N | \hat{J} | M - N - 1, N + 1 \rangle = J_{RP} \langle \Phi_{M-N}^N | \Phi_{M-N-1}^{N+1} \rangle.$$

REVIEWERS' COMMENTS:

Reviewer #1 (Remarks to the Author):

Having read the correspondence with referee two, I remain strongly in support of the publication of this paper. I fully agree with the authors response to the other referee; it seems many of the referee's objections are based on assumptions about what the manuscript says that do not match what one finds from a careful reading.

In most cases of such disagreement, I would suggest the disagreement might indicate there are points the authors would need to make more clear, to avoid such confusion occurring for other readers. However, in this case, the authors have already made such clarifications, and on re-reading the manuscript, I am convinced that the manuscript as it stands is clear at least to a physics audience.

Regarding one specific point of disagreement, I note that it was clear to me from the first version that the authors were refereeing to the specific effect of high frequency vibrational modes, and it was also always clear that the effect they describe depends only on the role of strong coupling in the final product state. The authors have not changed their story about this, they have only added wording to clarify the points they already made.

In summary, I reiterate my strong support for publication of this manuscript. The disagreement with referee two in fact makes me more persuaded of the significance of this manuscript: the manuscript makes an important point about how the eigenstate structure associated with high frequency modes and the thermal activation associated with the classical modes in the Marcus theory interact to enable strong matter-light coupling to change transition rates. The result is non-trivial, and I believe important, and I find the manuscript is clear about this main point.